# Chemical Modification of Chitosan for Removal of Pb(II) Ions from Aqueous Solutions

**DOI:** 10.3390/ma14247894

**Published:** 2021-12-20

**Authors:** Adriana Popa, Aurelia Visa, Bianca Maranescu, Iosif Hulka, Lavinia Lupa

**Affiliations:** 1Organic Chemistry Department, “CoriolanDragulescu” Institute of Chemistry, 24 Mihai Viteazul Blv, 300223 Timisoara, Romania; apopa@acad-icht.tm.edu.ro (A.P.); apascariu@yahoo.com (A.V.); 2Department of Biology-Chemistry, Faculty of Chemistry, Biology, Geography, West University Timisoara, 16 Pestalozzi Street, 300115 Timisoara, Romania; 3Research Institute for Renewable Energy, Politehnica University Timisoara, 138 Gavril Muzicescu, 300501 Timisoara, Romania; iosif.hulka@upt.ro; 4Faculty of Industrial Chemistry and Environmental Engineering, Politehnica University Timisoara, 6 Vasile Parvan Blv, 300223 Timisoara, Romania

**Keywords:** chitosan, aminophosphonic groups, modified chitosan-supported Ni(II) ions, lead adsorption

## Abstract

Biomacromolecule have a significant contribution to the adsorption of metal ions. Moreover, chitosan is one of the most studied biomacromolecule, which has shown a good performance in the field of wastewater treatment. In this context, a new adsorbent of the aminophosphonic modified chitosan-supported Ni(II) ions type was prepared from the naturally biopolymer, chitosan. In the first step, modified chitosan with aminophosphonic acid groups was prepared using the “one-pot” Kabachnik-Fields reaction. It was characterized by different techniques: FTIR, SEM/EDAX, TGA, and ^31^P-NMR. In the second step, the modified chitosan with aminophosphonic acid was impregnated with Ni(II) ions using the hydrothermal reaction at different values of pH (5, 6 and 7). The physical-chemical characteristics of final products (modified chitosan carrying aminophosphonic groups and Ni(II) ions) were investigated using FTIR, SEM images, EDAX spectra and thermogravimetric analysis. In this work, the most important objective was the investigation of the adsorbent performance of the chitosan modified with aminophosphonic groups and Ni(II) ions in the process of removing Pb(II) ions from aqueous solutions by studying the effect of pH, contact time, and Pb(II) ions concentration. For removal of Pb(II) ions from the aqueous solution, the batch adsorption method was used.

## 1. Introduction

Chitosan is derived from natural sources, such as insects’ skeletons, crustaceans and fungi that act to be biocompatible and biodegradable [1,2]. Chitosan polymers are semisynthetic derivatives of aminopolysaccharides with distinctive structures [3,4]. It is a polycationic polymer composed of two types of units: an N-acetyl-d-glucosamine unit together with a d-glucosamine unit, with the arrangement of these units depending on the proportion of acetylated fragments [5,6,7]. Chitosan has two hydroxyl groups and one amino group in its glycosidic residue [7]. Particular attention is paid to chitosan due to its very efficient biosorbent properties and low cost (compared to active carbon) and its high content of functional hydroxyl and amino groups [2,3].

This biopolymer is a smart substitute for other biomaterials because of its physicochemical characteristics: high reactivity, excellent chelation behavior, chemical permanence and high selectivity to pollutants [8,9,10].

Chitosan can be chemically modified to generate new biofunctional materials, as chemical modification changes the chitosan skeleton and ultimately brings new properties depending on the environment of the newly introduced unit. Changing chitosan by introducing phosphonic acid, phosphonate and aminophosphonic acid groups by synthesis of the phosphorylating agent with the chitosan amino groups is known to achieve increased chelating properties [1,2,3].

They have highly functionalized properties, and are widely used in biomedical and industrial domains [5]. Natural chitosan can be modified by various methods (both physical and chemical) in order to increase the adsorption capacity against different varieties of pollutants. Different forms of chitosan, for example: membranes, microspheres, gels, beads and films have been prepared and used to remove various pollutants from waste waters.

Chitosan and products derived from it have been presented in treatment processes of water for the elimination of metal ions such as: zinc(II), copper(II), cadmium(II), nickel(II), lead(III), and lead(II) ions [2,11,12], and for the removal of fluoride from waste waters [13,14,15,16,17]. The potential of chitosan in adsorption of heavy metals can be linked to: the high hydrophilicity on glucose units due to the high number of hydroxyl groups, the high reactivity of the additional functional groups and the flexible structure of the polymer chain [10].

Natural polysaccharide-type absorbents (e.g., chitosan or cellulose) are used to remove the Pb(II) ion, which has been shown to have good sorption properties [15,16,17]. Pollution of lead ions in wastewater is a serious problem. Therefore, we examined the ability of ChitPNi as an adsorbent for lead ions, expecting good adsorption properties.

In previous articles, we have reported the chemical modification of polymers [11,18]. That is a method of great interest because were obtained the efficient materials for environmental remediation such as: adsorbent material [11], antibacterial polymer [18], catalyst [19,20] and corrosion protection [21]. Obtaining chitosan derivatives by chemical modification is an excellent method exemplified by reviews in recent years [22,23,24].

The present paper mainly focuses on the chemical modification of chitosan with 1-phosphonopropyl groups (α-aminophosphonic acid groups) and nickel ions and its adsorption performance. The efficient synthesis of modified chitosan with 1-phosphonopropyl groups (α-aminophosphonic acid groups) and Ni(II) ions was analyzed by varying the reaction pH. The samples of both modified chitosan with α-aminophosphonic acid and modified chitosan with α-aminophosphonic acid and Ni(II) ions were investigated by Fourier Transform Infrared spectroscopy (FTIR), Scanning Electron Microscopy (SEM), and Energy Dispersive X-ray spectroscopy (EDAX), and Thermogravimetric Analysis (TG/DTA). The modified chitosan with α-aminophosphonic acid was analyzed by Nuclear Magnetic Resonance spectroscopy (^13^C-NMR and ^31^P-NMR). The adsorbent performance of the modified chitosan with α-aminophosphonic acid and Ni(II) ions was studied in the process of removing Pb(II) ions from aqueous solutions. The effects of pH, initial lead concentration, and contact times upon adsorption properties were investigated in detail.

## 2. Materials and Methods

### 2.1. Materials

Chitosan (the degree of N-deacetylation (%) of chitosan is 75–85% DD, the degree of N-acetylation (%) of chitosan is 15–25% DA, molecular weight is 190,000–310,000 DA, Sigma-Aldrich, Darmstadt, Germany), phosphorus acid 99% (Sigma-Aldrich), propionaldehyde 97% (Sigma-Aldrich), acetone (Chimreactiv, p.a), nickel acetate(II) tetrahydrate (Ni(OCOCH_3_)_2_·4H_2_O, purum p.a., ≥99.9%, Sigma-Aldrich) were used for preparation of modified chitosan. All chemicals used in our work were of analytical reagent grade and were used without further purification.

### 2.2. The Chemical Modification of Chitosan with Aminophosphonic Groups Using the “One-Pot” Kabachnik-Fields Reaction

Six g of chitosan powder was dissolved in aqueous acetic acid 2% (*w*/*v*) (100 mL), and, after complete dissolution was placed in a round bottomed flask. The reaction mixture was prepared by adding both the phosphorous acid (2 g) dissolved in distilled water and propionaldehyde (2.5 mL) in dropwise over the solution of chitosan at the temperature of 50 °C. The reaction of chemical modification of chitosan was remained at 70 °C, for time of 24 h. Finally, the product was precipitated in acetone, filtrated and the obtained product (ChitP) was dried in an oven at temperature of 30 °C for 24 h.

### 2.3. The Impregnation with Ni(II) Ions of Chemically Modified Chitosan

Next, 1.2 g of modified chitosan (ChitP) (phosphorus content 14.43% by weight) was placed in an Erlenmayer flask and it was dissolved in 50 mL of 1% glacial acetic acid solution under magnetic stirring for 12 h until the complete dissolution occurred. In another glass, the nickel acetate (II) tetrahydrate (Ni(OCOCH_3_)_2_·4H_2_O) was dissolved in 100 mL of distilled water and 0.5 g of urea was added under stirring until a clear solution was formed. Under the reaction conditions, urea is slowly hydrolyzed with the release of NH3 and allowing the pH raising in a homogeneous manner. The molar ratio used was 1:1 = Ni(II) ions: phosphorus contained in the aminophosphonic group. The clear solutions were mixed and the pH value of the resulting mixture was 4.15. The method described above was repeated 2 times to obtain 3 glasses of samples with solutions of modified chitosan carrying aminophosphonic groups and Ni(II) ions. Afterwards, the final pH of the solutions was adjusted to different values 5, 6, and 7 by adding 0.1N NaOH. The solutions thus obtained were placed in a water bath at 80 °C for 60 h. The precipitate formed was filtered, washed with distilled water and the final products (ChitPNi) were dried in an oven at 50 °C for 48 h.

### 2.4. Characterization

The FTIR spectrum (performed on KBr tables) of the obtained material was recorded using a JASCO-FT/IR-4200 spectrophotometer (JASCO, Tokyo, Japan). The surface morphology of the investigated samples was studied by Scanning Electron Microscopy (SEM) using a Quanta FEG Microscope equipped with EDAX ZAF quantifier—(FEI Company, Hillsboro, OR, USA). The thermal stability of the sample was investigated by differential thermography (TG-DTA) by using a Mettler-Toledo machine (Greifensee, Switzerland), in the temperature range 25–700 °C, at a heating rate of 10 °C/min, under a nitrogen and air atmosphere.

The ^31^P NMR spectrum of modified chitosan with aminophosphonic acid (ChitP) was recorded on a Bruker DRX 400 MHz spectrometer (Bruker Corporation, Billerica, MA, USA), in CD_3_COOD/D_2_O at 298 K. The sample was dissolved in 2 wt. CD_3_COOD/D_2_O at a 20 mg mL^−1^ concentration, and the NMR tube was kept at 60 °C to dissolve the polymer % in solution for 24 h. The pH of the solutions was measured via a CRISON MultiMeter MM41 (Crison Instruments, Barcelona, Spain) fitted with a glass electrode, which had been calibrated using various buffer solutions. Lead ion concentrations were determined using a Varian SpectrAA 280 Fast Sequential Atomic Absorption Spectrometer (Varian, Duisburg, Germany). For batch experiments, a Julabo SW23 shaker bath (Labortechnik GmbH, Sellbach, Germany) was used. All the solutions used in this experiment were prepared with distillated water.

X-ray photoelectron spectroscopy (XPS) was performed on a KRATOSAxisNova (KratosAnalytical, Manchester, UK), using AlK radiation, with a current of 20 mA and a voltage of 15 kV (300 W). The XPS spectrum for the tested sample was made in the binding energy range −10–1200 eV, using pass energy of 160 eV with a resolution of 1 eV. The XPS spectrum for all identified elements (N, P, O, C, Ni) was performed using a pass energy of 20 eV and a step size of 0.1 eV. Vision Processing software (Vision2 software, Version 2.2.10) was used for data analysis.

The X-ray diffractograms of ChitP and ChitPNi were recorded using a Rigaku Ultima IV X-ray diffractometer (Rigaku Europe SE—Neu-Isenburg, Germany) (40 kV, 40 mA) with Cu_Kα_ radiation.

### 2.5. Adsorption Studies of the Modified Chitosan with Aminophosphonic Acid Groups and Ni(II) Ions (ChitPNi)

The chitosan with aminophosphonic acid groups and Ni(II) ions (ChitPNi), which was obtained and characterized was used as the adsorbent material in the treatment of lead containing aqueous solutions. Adsorption experiments are conducted using under continuous stirring at a speed of 200 rpm, in batch mode, using a Julabo SW23 shaker bath. A stock solution of 1 g L^−1^ Pb(NO_3_)_2_ was used for the whole process. The aqueous solutions containing different amounts of lead ions were obtained through adequate dilution of the stock solution using distilled water. During the adsorption process, the initial pH of the lead containing aqueous solutions was adjusted around the value pH = 5. At this pH, the obtained modified adsorbent presents the best characteristics, and also according to a literature survey, the best pH value to use in the adsorption of lead ions onto various chitosan derivatives is in the range: 4.5–5.5 [25,26]. By using the adsorbent in this condition, leaching of the nickel ions from the functionalized chitosan is avoided.

The adsorption process was discussed from the kinetic and equilibrium point of view, studying the modified chitosan adsorption performance, function of the stirring time, and function of the initial lead concentrations. The chitosan adsorption performance was expressed as the quantity of lead ions adsorbed onto one g of functionalized chitosan, *q_m_* in Equation (1).
(1)$$q_{e}=\frac{\left(C_{0}-C_{e}\right)\cdot V}{m}$$
where: ***q_e_*** is the quantity of lead ions adsorbed in (mg Pb) per 1 g of adsorbent of studied adsorbent; ***C*_0_** and ***C_e_*** represent the initial and equilibrium lead concentrations of the aqueous solutions (mg L^−1^). ***V*** is the volume of the lead containing aqueous solutions (L) and ***m*** is the mass of the functionalized chitosan (g) used in the experiments.

### 2.6. Adsorption Theory Background

In order to determine the optimum operational conditions necessary to treat lead containing aqueous solutions at full scale, it is necessary to examine the results of the adsorption experiments from the kinetic and equilibrium point of view.

#### 2.6.1. Kinetic Studies

The studies on the influence of stirring time upon the adsorption performance have been used to determine the time necessary to establish the equilibrium between modified chitosan with aminophosphonic acid groups and Ni(II) ions (ChitPNi) and the adsorbate (Pb(II)). The experimental results can be modeled with different kinetic isotherms in order to establish the rate limiting step and the type of adsorption mechanism [27,28,29]. In this paper, three kinetic models are discussed: pseudo-first order, pseudo-second order and intra particle diffusion.

The pseudo-first order model is expressed in its linear form, according to Equation (2) [27,28,29]:(2)$$\ln\left(q_{e}-q_{t}\right)=\ln q_{t}-k_{1}\cdot t$$
where: ***q_e_*** and ***q_t_*** represent the quantity of lead adsorption onto 1 g of (ChitPNi) at equilibrium and after the treatment time, ***t***; ***k*_1_** represent the rate constant specific for the adsorption process of lead ions onto modifieds chitosan with aminophosphonic acid groups and Ni(II) ions (ChitPNi) (min^−1^). The rate constant and the equilibrium adsorption capacity were determined from the slope and intercept of ***ln***(***q_e_*** − ***q_t_***) versus time of stirring.

If the process of lead adsorption onto the modified chitosan (ChitPNi) is controlled by the chemisorption rate, then the experimental data are well modeled by the pseudo-second order kinetic model in its linear form, in Equation (3) [27,28,29]:(3)$$\frac{t}{q_{t}}=\frac{1}{k_{2}\cdot q_{e}^{2}}+\frac{t}{q_{e}}$$
where: ***q_e_***, ***q_t_*** and t have the same significance as shown above, and ***k*_2_** is the rate constant of the pseudo-second order kinetic model. These parameters could be determined from the slope and intercept of the linear plot between ***t***/***q_t_*** and stirring time.

The rate limiting step in the process of adsorption of lead ions onto (ChitPNi) could be established from the intraparticle diffusion experimental data model, in Equation (4) [29,30].
(4)$$q_{t}=k_{in}\cdot t^{1/2}+C$$
where: ***k_in_*** represents the rate constant of the intraparticle diffusion model.

#### 2.6.2. Equilibrium Studies

Equilibrium studies have been conducted to determine the interactions between the modified chitosan with aminophosphonic acid groups and Ni(II) ions (ChitPNi) surface and the lead ions. Additionally, the maximum adsorption capacity developed by the modified chitosan (ChitPNi) in the adsorption of lead ions from aqueous solutions was studied. For this purpose, four isotherm models, based on two parameters, were applied to model the equilibrium data obtained experimentally: Langmuir, Freundlich, Temkin and Dubinin–Radushkevich.

The linearized Langmuir isotherm represented by the Equation (5) suggests that the lead adsorption onto modified (ChitPNi) chitosan occurs as a monolayer covering the adsorbent surface [31,32,33].
(5)$$\frac{C_{e}}{q_{e}}=\frac{1}{q_{m}\cdot K_{L}}+\frac{C_{e}}{q_{m}}$$
where: ***q_m_*** represents the maximum adsorption capacity of the modified (ChitPNi) chitosan and ***K_L_*** represents the Langmuir constant. These two parameters can be determined from the linear plot of ***C_e_***/***q_e_*** versus ***C_e_***.

The affinity between the modified (ChitPNi) chitosan and the lead ions can be obtained by fitting the data to the linearized form of the Freundlich isotherm, in Equation (6) [26,27,28]:(6)$$\ln\left(q_{e}\right)=\ln(k_{F})+\frac{1}{n}\ln(C_{e})$$
where: **1**/***n*** and the Freundlich constant ***k_F_*** represent the Freundlich isotherms parameters and can be determined from the plot of ***ln***(***q_e_***) versus ***ln***(***C_e_***).

If the surface of the adsorbent is heterogeneous, the equilibrium data will present a good fit for the linearized form of the Temkin isotherm model, which is given by Equation (7), suggesting that the heat of adsorption decreases during the adsorption process [31,32,33].
(7)$$q_{e}=\frac{R\cdot T}{b}\ln k_{T}+\frac{R\cdot T}{b}\ln(C_{e})$$
where: ***k_T_*** is the equilibrium binding constant, and ***b*** is related to the heat of adsorption, which were determined from the linear plot between ***q_e_*** and ln(***C_e_***).

The modelling of the experimental data according to the linearized form of the Dubinin-Radushkevich (Equation (8)) lead to the obtaining of the mean sorption energy E, which is given by Equation (9), which indicates whether the type of adsorption process is physical or chemical [31,32,33].
(8)$$\ln\left(q_{e}\right)=\ln(q_{m})-K_{ad}\cdot \varepsilon^{2}$$
(9)$$E=\frac{1}{\sqrt{2K_{ad}}}$$
where: ***q_m_*** and ***K_ad_*** can be determined from the slope and intercept of the linear plot ln(***q_e_***) = f(***ε***)^2^.

For all the isotherm modeling used in the equilibrium studies, a correlation coefficient, ***R***^2^, had to be obtained. This coefficient had to be close to unity, in order to suggest that the given equations fit the experimental data. In addition to these correlation coefficients, for a better fit (even when the values of ***R***^2^ are close to each other) three error functions were analyzed for each adsorption isotherm: chi-square analysis (***χ*^2^**)—Equation (10), root mean square error (RMSE)—Equation (11) and the sum of the square of the errors (ERRSQ)—Equation (12). The lower the errors analysis, the better the fits are [34].
(10)$$\chi^{2}=\sum_{i=1}^{N}\left[\frac{{\left(q_{e,\;exp}-q_{e,\;calc}\right)}^{2}}{q_{e,calc}}\right]$$
(11)$$\mathrm{RMSE}=\sqrt{\frac{1}{N}\sum_{i=1}^{N}{\left(q_{e,\;exp}-q_{e,\;calc}\right)}^{2}}$$
(12)$$\mathrm{ERRSQ}=\sum_{i=1}^{N}{\left(q_{e,\;calc}-q_{e,exp}\right)}^{2}$$
where: each term has its previously mentioned meaning and ***N*** represent the number of data items.

## 3. Results and Discussion

### 3.1. Adsorbent Characterization

Figure 1 presents the modified chitosan with aminophosphonic acid (ChitP).

Figure 1 presents for comparison the FTIR bands both of (ChitP) and (ChitPNi pH 5).

The recorded FTIR spectrum of (ChitPNi) pH 5 (see Figure 1) indicates the presence of a 1050 cm^−1^ band assigned to the –P-O-H group. The N-H stretch strips and band appear at 1655 and 1560 cm^−1^. A strong absorption is observed at 3430 cm^−1^, corresponding to the tensile vibration of the hydroxyl groups in the water coordinated with the Ni atom superimposed with the tensile vibration N-H. At 1420 cm^−1^, it was assigned a band for OH bending vibration. The band at 1540 cm^−1^ is due to vibrations of the carboxylate ion. The absence of bands in the range 2700–2560 cms^−1^, corresponding to the P-O vibration of the P-OH acid groups, confirms the formation (ChitPNi) [35,36]. These dates are in line with the proposed structures of functionalized chitosan (ChitPNi).

The FTIR confirms the existence of phosphorus on the polymeric support, and thus the aminophosphonic group is active and allows the binding of Ni(II) ions. The syntheses were recorded at different pH because the crystallinity of the product is highest at pH 5 [37], it was chosen for our study.

The characteristic chemical shifts for ChitP in ^13^C NMR appear at 18.41, 22.10, 55.77, 77.56, 59.93, 70.00, 74.73, 76.27 and 97.48 ppm and assigned to CH_3_, CH_2_, C_6_, C_2_, CH, C_5_, C_3_, C_4_ and C_1_ carbons, respectively.

The ^31^P-NMR spectrum of the raw material (ChitP) displayed only one signal at 2.858 ppm. This peak is generally expected for phosphonic group [36,38]. The ^31^P-NMR spectrum confirms the successful incorporation of the phosphorous group is chemically bonded to the chitosan.

Figure 2 presents a semi-quantitative EDAX analysis of chitosan modified with aminophosphonic groups impregnated with Ni(II) ions where the presence of nickel can be seen in the spectrum. The impurities recorded by the device (<1%) were not shown in Table 1.

The X-ray photoelectron spectroscopy (XPS) method was described in a previous article [39]. In the analysis of the XPS spectrum in Figure 3, we obtained elements: C, N, O, P and Ni. From XPS spectrum was observed that ChitPNi exhibit for phosphorus and Ni ions comparable with the one presented in the literature [40,41,42]. Additionally, the values for the phosphorus, nitrogen and nickel content are given in Figure 3.

Termogravimetric analysis (TG and DTA) was carried out from 25 to 700 °C under a nitrogen and air flow, in 10 °C min^−1^ heating step.

The thermal stability of the raw material (ChitP) in nitrogen and of the final products (ChitPNi at pH = 5, 6 and 7) in nitrogen is presented in Figure 4. From the TG curves, it can be observed that the ChitP sample has a mass loss of 61.07% compared to ChitPNi (pH = 5) with a mass loss of 44.62%. For (ChitPNi) products at pH = 6 and 7, the mass losses of 47.56% and 56.26%, respectively, were obtained. Thus, the sample (ChitPNi) to pH 5 has a higher stability, we have further studied it in order to use for recovery of lead ions.

Figure 5 shows the TG and DTA curves for ChitPNi (pH 5). The thermogravimetric analysis of the TG/DTA diagrams recorded in nitrogen for the (ChitPNi) sample at pH = 5 (Figure 4) shows three distinct mass losses [40]. The first loss between 50–180 °C was a mass loss of ~15.37%, which was assigned to the evaporation of the physically adsorbed water and the organic compounds from the acetate decomposition [40]. In the second stage between 190–380 °C the mass loss of ~44.62% was associated with the decomposition of organic compounds, the degradation of the N-C-P bond between chitosan and the aminophosphonic group, and the formation of nickel [36]. The third step in the temperature range 390–700 °C shows a mass loss of ~6.08%, which is attributed to the degradation of the complex P-O structure together with the degradation of the carbohydrate nucleus [36,40].

From the TG/DTA diagrams of the (ChitPNi) sample at pH = 5 (Figure 6) in an atmosphere of air, an endothermic effect is observed in the range 60–150 °C. The first mass loss of 13.60% is a result of the evaporation process of the water in the sample, followed by a sequence of exothermic effects in the range 180–700 °C due to the oxidative degradation of the sample. At 320 °C, NiO is obtained [36,40], and at 358 °C, P_2_O_5_ is detached [43]. The mass loss of the oxidative degradation step was 42.62%, and the total loss of the (ChiPNi) sample is 56.21%.

The RX-diffractograms of ChitP and ChitPNi samples are presented in Figure 7. The X-ray diffraction of ChitP exhibit very broad peak at 2θ = 20°, which is specific for chitosan-based samples. The XRD pattern obtained for modified sample, ChitPNi shows that the specific peak of chitosan is doubled, with another peak at 2θ = 26.48°, which corresponds to γ-NiOOH, and depicts a peak at 2θ = 36.48° corresponding to the miller indices (111). These results suggest the interactions between Ni and chitosan, which lead to obtaining an amorphous sample with a higher specific surface area, which is expected to develop a higher adsorption capacity.

### 3.2. Adsorbent (ChitPNipH5) Characterization after Pb(II) Ions Adsorption

The SEM image and EDAX spectra after Pb(II) ions adsorption onto modified chitosan (ChitPNi) is presented in Figure 8. It can be observed that after Pb(II) ions adsorption, the surface of the modified chitosan is more compact and rigorous, suggesting that the lead ions are adsorbed onto its surface, filling all the pores and linking all the active sites. The Pb(II) ions adsorption is confirmed by the characteristic peak, which appears in the EDAX spectra of the sample. The elements quantification is presented in Table 2.

### 3.3. Kinetic Studies

The equilibrium between the modified chitosan (ChitPNi) and adsorbed Pb(II) ions is achieved after 120 min of contact, as can be noticed from Figure 9, which represents the adsorption capacity in relation to stirring time.

The adsorption capacity increased quickly in the first half, and after this time it increased more slowly, achieving the equilibrium at 120 min. The equilibrium is achieved more quickly than in the other reported adsorption processes with lead ions, for example, onto chitosan-grafted-poly acrylic acid (t_e_ = 240 min) [33] and, respectively, more than 5 h onto chitosan impregnated granular activated carbon [26]. For example, with the adsorption of lead onto chitosan-grafted-poly acrylic acid, equilibrium was achieved after 4 h, and with lead and chitosan impregnated granular activated carbon, the time was more than 5 h.

The linear plots of the kinetic model studied are presented in Figure 10. The kinetic parameters obtained from the slope and intercept of these plots, together with the correlation coefficients, are summarized in Table 3.

The kinetic parameters presented in Table 3 for the pseudo-first order and pseudo-second order kinetic models were compared, and it was noticed that the adsorption of Pb(II)ions onto modified chitosan (ChitPNi) is best described by the second model. The highest correlation coefficient (R^2^ = 0.9981) is obtained for the second order correlation model and the adsorption capacity achieved by modified chitosan in the removal of Pb(II)ions is closest to that calculated one. Therefore, it would seem that the removal of Pb(II)ions from aqueous solutions is due to chemisorption processes [27,28,29].

This chemisorption process is a complex one, which takes place in two steps, according to the intraparticle diffusion plot (Figure 10c). It is obvious that the intraparticle diffusion is not the rate-limiting step for adsorption of Pb(II) onto the modified chitosan (ChitPNi). In the first 10 min, the Pb(II) diffuses through the solution until it reaches the (ChitPNi) chitosan surface, then the rate limiting step is controlled by the adsorption of the Pb(II) inside the (ChitPNi) chitosan particles [29,30].

### 3.4. Equilibrium Studies

The maximum adsorption capacity of the ChitP and modified chitosan, ChitPNi developed for the removal of Pb(II) can be experimentally determined from the equilibrium isotherm presented in Figure 11. The maximum adsorption capacity obtained experimentally for Pb(II) removal is of 48.1 mg Pb(II) g^−1^ of modified chitosan (ChitPNi) and 21 mg/g for ChitP. It is observed that the modification of ChitP with Ni leads to a doubling of the maximum adsorption capacity of the studied materials. As the modified chitosan developed a high adsorption capacity, the forward experiments were made only with the newly obtained adsorbent materials.

The linear plots of the equilibrium isotherm studied are presented in Figure 12. The equilibrium parameters, together with the correlation coefficients and the error values, are listed in Table 4.

From the data listed in Table 4, it can be observed that the Pb(II)ions adsorption onto modified chitosan (ChitPNi) is best described by the Langmuir isotherm. In this case, the correlation coefficient is close to 1, and the values for error function are low. Furthermore, a maximum adsorption capacity for Pb(II)ions adsorption onto (ChitPNi) of 50.3 mg g^−1^ was obtained, which is close to that obtained experimentally. For the entire concentration range the *R_L_* values are in the frame of 0–1 range indicating a favorable adsorption of Pb(II) ions onto modified chitosan (ChitPNi). The adsorption of lead ions onto the surface of the (ChitPNi) in a mono layer involves a lot of surface energy and because of the great bonding between the Pb(II) ions and modified chitosan (ChitPNi), the value for *K_L_* was high (0.150 L mg^−1^) [31]. The Freundlich isotherm presents a correlation coefficient *R*^2^ > 0.9, but values for error function were also high, which limits the usefulness of this adsorption isotherm in describing the data for the adsorption of Pb(II) ions onto (ChitPNi). Even if the 1/n value indicates a high affinity of modified chitosan for Pb(II) ions the value obtained for *K_F_* = 7.995 mg g^−1^ suggests a low quantity of Pb(II) ions adsorbed onto modified chitosan which contradicts the results obtained experimentally [29,30,31,32]. As with the Langmuir isotherm model, the Temkin isotherm gave good results, suggesting that the surface of the (ChitPNi) sample is of the heterogenous type. The Dubinin–Radushkevich model resulted in the lowest values for *R*^2^ and for the error functions. The value of the mean sorption energy *E* = 1.44 kJ mol^−1^, obtained from the Dubinin–Radushkevich modeling suggests physiosorption of Pb(II) ions onto (ChitPNi) [26].

The maximum adsorption capacity acquired from the Langmuir isotherm for the removal of Pb(II) ions onto modified chitosan (ChitPNi) was compared with the maximum adsorption capacities reported by other researchers in literature on adsorption capacities of other derivatives of chitosan, or of other biosorbents (Table 5).

It can be observed that the modified chitosan (ChitPNi) studied in this paper presents a good adsorption efficiency in the removal of Pb(II) ions compared with similar materials presented in the specialty literature.

For the recovery of lead ions from the ChitPNi surface, a solution of 1 M HCl was used, and the samples were mixed for 15 min. After the regeneration process, the phases were separated and the recycled adsorbent, ChitPNi, was used in other adsorption processes and the extracted lead ions from the solution were determined (Figure 2). Additionally, the content of nickel ions in solution after lead adsorption was investigated, by atomic absorption spectrometry. Any nickel content was identified in solution, and this means that there is now an ionic exchange between nickel and lead ions during the adsorption process. This aspect can be also be observed on the semi-quantitative EDAX analysis (Table 1 and Table 2).

## 4. Conclusions

In our study, the modified chitosan with aminophosphonic groups and Ni(II) ions (ChitPNi) was prepared as a novel adsorbent material to remove Pb(II) from water. The (ChitPNi) adsorbent material was obtained by using chitosan modified with aminophosphonic groups (ChitP) through the hydrothermal reaction. Additionally, the raw material, ChitP, was previously obtained by the chemical modification reaction of chitosan with phosphorous acid and propionaldehyde. The achieved results of the FTIR specroscopy, SEM, EDAX and TG analysis and the ^13^C NMR and ^31^P NMR spectra of (ChitP) and of the RX diffraction show that the modified chitosan with aminophosphonic groups (ChitP) and the modified chitosan with aminophosphonic groups Ni(II) ions (ChitPNi) were obtained. Through modification of ChitP with Ni ions, an adsorbent material with higher active site and higher specific surface area is obtained, which leads to the development of a higher adsorption capacity (*q*_max_ = 50.25 mg/g) in the removal process of Pb ions from aqueous solutions compared with nickel free ChitP (*q*_max_ = 21 mg/g).

The kinetic and equilibrium studies confirmed the fact that the Pb(II) adsorption corresponds to a complex process involving both physisorption of the metal ions onto the pores of the adsorbent and also chemisorption as the metal ions link with the active sites of the (ChitPNi) illustrated into the Pb^2+^ ions removal over ChitPNi scheme. This modification of chitosan leads to the obtaining of a potential adsorbent with good performance in treating water with lead content.

## Data Availability

Data are contained within the article.

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
