# Peer review of "Chemical Modification of Chitosan for Removal of Pb(II) Ions from Aqueous Solutions"

_materials, 2021, doi:10.3390/ma14247894_

Round 1
Reviewer 1 Report
The paper by Adriana Popa et al. is focused on the preparation and characterization of a bioadsorbent for Pb(II) ions based on N-(1-phosphonopropyl)chitosan. The article is written and prepared very carelessly, and therefore it is vague. The paper could have some scientific contribution, but it has to be significantly improved and completely rewritten. At this point, I am forced to reject the paper for the following main reasons:
- First, English usage is often poor. There are too many English errors and wrong constructions, which sometimes drastically affect the scientific meaning, making the paper difficult to read and understand. A native English speaker with a scientific background should carefully revise the manuscript before its resubmission.
- Authors often misuse the terminology. For example, they believe that they obtained chitosan modified with an aminophosphonic group. However, the amino group belongs to chitosan, so it is modified with 1-phosphonopropyl group.
- The Introduction section is quite weak. The authors do not provide sufficient background to let the reader understand the basic topic and what the study aims were. The introduction does not contain a problem statement, objective or research question, motivation, or reasons behind the study.
- Some of the statements in the paper are incorrect. For example, the authors stated (lines 44-47) that chitosan, when chemically modified, “retains its initial physicochemical properties and biochemical properties, but ultimately brings new properties depending on the environment of the new unit introduced.” It is hard to agree with this statement because any chemical modification of the polymer (chitosan) will inevitably lead to a change in its physicochemical and biological properties.
- Lines 63-66: I understand the desire of the authors to promote their own publications; however, the scientific community is opposing the massive self-citations. I would recommend the authors revise the references and leave only those needed for discussion (most of the self-cited papers are not relevant to the topic of this paper). Nature published an article discussing this issue https://www.nature.com/articles/d41586-019-02479-7.
- The chitosan must be thoroughly characterized regarding its molecular weight (by viscometry, light scattering, or size exclusion chromatography) and the degree of deacetylation (by NMR, IR, elemental analysis, or titration). The properties of chitosan are very dependent on these parameters; therefore, the wide ranges of values provided by the manufacturer are insufficient.
- Line 103: What is the role of urea in this process?
- Section 3.1. Adsorbent characterization: Characterization of the ChitP sample using IR spectroscopy only is obviously insufficient. It is necessary to prove the modification of chitosan by the 1-phosphonopropyl group (e.g., by 1H, 13C NMR), to determine the degree of substitution (e.g., by 1H NMR), the degree of acetylation (e.g., by 1H NMR), and the molecular mass of ChitP (e.g., by SEC-MALS).
- The quality of FTIR spectra (Fig. 1) is poor as well as the description and discussion of the spectra (lines 242-254) are very confusing and must be revised entirely. Incorrect assignment of absorption bands (e.g., C-O from acetylated moieties of chitosan at 1350-1300 cm-1). The possible presence of acetate is not taken into account (although they mention the presence of acetic acid in the samples when describing the TG/DTA experiments, line 283).
- Lines 254-256: How did the authors reach this conclusion without examining the degree of crystallinity?
- Line 258: What do the authors mean by “31P functionalities”?
- Line 298: How can NiO be removed at 320 °C as its melting point is 1.955 °C?
- Lines 397-400: Does nickel release during the regeneration of 0.1 M HCl adsorbent? If not, what would be the reason?
- The discussion of the results of the paper is also rather weak. I still do not comprehend the role of nickel ions in the adsorbent. The authors claimed that there is no competitive sorption of lead, leading to displacement of nickel from the sorbent. If so, maybe nickel is not needed in the sorbent at all? To resolve this question, I would like to see in the paper a sorption experiment performed on a nickel-free sample (ChitP).
Author Response
Dear Reviewer, thank you for your suggestion and comments, we improved our manuscript according with you recommendation and we hope that in this format this could be accepted for publication.
The paper by Adriana Popa et al. is focused on the preparation and characterization of a bio adsorbent for Pb(II) ions based on N-(1-phosphonopropyl)chitosan. The article is written and prepared very carelessly, and therefore it is vague. The paper could have some scientific contribution, but it has to be significantly improved and completely rewritten. At this point, I am forced to reject the paper for the following main reasons:
- First, English usage is often poor. There are too many English errors and wrong constructions, which sometimes drastically affect the scientific meaning, making the paper difficult to read and understand. A native English speaker with a scientific background should carefully revise the manuscript before its resubmission.
Answer 1: Thank you for your suggestion. We checked English and turned to a native English speaker
- Authors often misuse the terminology. For example, they believe that they obtained chitosan modified with an aminophosphonic group. However, the amino group belongs to chitosan, so it is modified with 1-phosphonopropyl group.
Answer 2: Thank you for the recommendation, yes indeed your suggestion is welcome. We made a chemical modification reaction to the existing amino group on chitosan with propionaldehyde and phosphorous acid (obtaining 1-propyl-phosphonic acid groups link with amino groups). We will respond to your recommendation in the introduction. We also want to suggest a reference that confirms that the name chitosan functionalized with aminophosphonate is usually used in articles:
- Asmaa S. Morshedy, Ahmed A. Galhoum, Abdel Aleem H. Abdel Aleem, Mohamed T. Shehab El-din , Dina M. Okaba, Mohsen.S. Mostafa, Hamed I. Mira, Zhen Yang, Ibrahim E.T.- El-Sayed, Functionalized aminophosphonate chitosan-magnetic nanocomposites for Cd(II) removal from aqueous solutions: Performance and mechanisms of sorption, Applied Surface Science. 2021, 561, 150069, 10.1016/j.apsusc.2021.150069
- Imam, E.A., El-Tantawy El-Sayed, I., Mahfouz, M.G., Tolba, A. A., Akashi, T., Galhoum, A.A., Guibal, E., Synthesis of α-aminophosphonate functionalized chitosan sorbents: effect of methyl vs phenyl group on uranium sorption, Chem. Eng. J. 2018, 352, 1022-1034, doi:10.1016/j.cej.2018.06.003
We inserted in the introduction “1-phosphonopropyl group” as mentioned in the following paragraph and we have kept the specification of "α-aminophosphonic acid" in the whole article as the name of the new formed group.
Lines 62-64:
“The present paper mainly focuses on the chemical modification of chitosan with 1-phosphonopropyl groups (α-aminophosphonic acid groups) and nickel ions and its adsorption performance. The efficient synthesis of modified chitosan with 1-phosphonopropyl groups (α-aminophosphonic acid groups) and Ni(II) ions was analyzed by varying the reaction pH.”
- The Introduction section is quite weak. The authors do not provide sufficient background to let the reader understand the basic topic and what the study aims were. The introduction does not contain a problem statement, objective or research question, motivation, or reasons behind the study.
Answer 3: Thank you for your suggestion. The introduction was improved and new paragraph with the references [15-17] about the sorption of Pb ions was introduced.
“Natural polysaccharide-type absorbents (eg chitosan or cellulose) are used to remove the Pb (II) ion, which has been shown to have good sorption properties [15-17]. Pollution of lead ions in wastewater is a serious problem. Therefore, we examined the ability of ChitPNi as an adsorbent for lead ions, expecting good adsorption properties.”
- Jin, X., Xiang, Z., Liu, Q., Chen, Y., Lu, F., Polyethyleneimine-bacterial cellulose bioadsorbent for effective removal of copper and lead ions from aqueous solution, Bioresour. Technol. 2017, 244, 844-849, doi:10.1016/j.biortech.2017.08.072
- Jiao Lu, Ru-Na Jin, Chao Liu, Yan-Fei Wang, Xiao-kun Ouyang, Magnetic carboxylated cellulose nanocrystals as adsorbent for the removal of Pb(II) from aqueous solution, Int. J. Biol. Macromol. 2016, 93, 547-556 doi:10.1016/j.ijbiomac.2016.09.004 0141-8130
- Kumar, R., Sharma, R.Kr., Synthesis and characterization of cellulose based adsorbents for removal of Ni(II), Cu(II) and Pb(II) ions from aqueous solutions, React. Funct. Polym. 2019, 140, 82-92, doi:10.1016/j.reactfunctpolym.2019.04.014
- Some of the statements in the paper are incorrect. For example, the authors stated (lines 44-47) that chitosan, when chemically modified, “retains its initial physicochemical properties and biochemical properties, but ultimately brings new properties depending on the environment of the new unit introduced.” It is hard to agree with this statement because any chemical modification of the polymer (chitosan) will inevitably lead to a change in its physicochemical and biological properties.
Answer 4: Thank you for your very useful observation. Line 44-47 skip phrases were deleted: Chitosan may be chemically modified to generate new biofunctional materials, since modification does not change the basic skeleton of chitosan, and it retains its initial physicochemical properties and biochemical properties, but ultimately brings new properties depending on the environment of the new unit introduced.
- We replace it with a new paragraph:
“Chitosan can be chemically modified to generate new bio functional materials, as chemical modification changes the chitosan skeleton and ultimately brings new properties depending on the environment of the newly introduced unit.”
- Lines 63-66: I understand the desire of the authors to promote their own publications; however, the scientific community is opposing the massive self-citations. I would recommend the authors revise the references and leave only those needed for discussion (most of the self-cited papers are not relevant to the topic of this paper). Nature published an article discussing this issue https://www.nature.com/articles/d41586-019-02479-7.
Answer 5: Thank you for your kindness in suggesting a reorganization of the bibliography.
This paragraph has been reorganized and references have been made with a change in the order of settlement.
Own publications were deleted, and new references were introduced.
“In previous articles, we have reported the chemical modification of polymers [11,18]. That is a method of great interest because were obtained the efficient materials for environmental remediation such as: adsorbent material [11], antibacterial polymer [18], catalyst [19,20] and corrosion protection [21]. Obtaining chitosan derivatives by chemical modification is an excellent method exemplified by reviews in recent years [22-24].”
- Popa, A.; Ilia, G.; Iliescu, S.; Dehelean, G.; Pascariu, A.; Bora, A.; Davidescu, C.M. Mixed quaternary ammonium and phosphonium salts bound to macromolecular supports for removal bacteria from water. Mol. Cryst. Liq. Cryst. 2004, 418, 923- 931. DOI: 10.1080=15421400490479280.
- Popa, A.; Parvulescu, V.; Tablet, C.; Ilia, G.; Iliescu, S.; Pascariu, A. Heterogeneous catalysts obtained by incorporation of polymer-supported phosphonates into silica used in oxidation reactions. Polym. Bull. 2008, 60, 149–158, DOI:10.1007/s00289-007-0844-z.
- Lazar, M.M., Dinu, I.A., Silion, M.; Dragan, E.S.; Dinu, M.V. Could the porous chitosan-based composite materials have a chance to a “NEW LIFE” after Cu(II) ion binding?, Int. J. Biol. Macromol. 2019, 131, 134–146. DOI:10.1016/j.ijbiomac.2019.03.055.
- Maranescu, B., Plesu, N.; Visa, A. Phosphonic acid vs phosphonate metal organic framework influence on mild steel corrosion protection, Appl. Surf. Sci. 2019, (497), 143734. https://doi.org/10.1016/j.apsusc.2019.143734
- Sahee, I.O., Oh,W.D., Suah, F.B.M., Chitosan modifications for adsorption of pollutants- A review, J. Hazard. Mater. 2021, 408, 12488, doi.org/10.1016/j.jhazmat.2020.124889
- Abd El-Hack, M.E., El-Saadony, M.T., Shafi, M.E., Zabermawi, N.M., Arif, M., Batiha, G. E., Khafaga, A.F., Abd El-Hakim, Y.M., Al-Sagheer, A.A., Antimicrobial and antioxidant properties of chitosan and its derivatives and their applications: A review. Int. J. Biol. Macromol. 2020, 164, 2726-2744,doi:10.1016/j.ijbiomac.2020.08.153
- Wang, W., Meng, Q., Li, Q., Liu, J., Zhou, M., Jin, Z., Zhao, K., Chitosan derivatives and their application in biomedicine, Int. J. Mol. Sci. 2020, 21, 487, doi:10.3390/ijms21020487
- The chitosan must be thoroughly characterized regarding its molecular weight (by viscometry, light scattering, or size exclusion chromatography) and the degree of deacetylation (by NMR, IR, elemental analysis, or titration). The properties of chitosan are very dependent on these parameters; therefore, the wide ranges of values provided by the manufacturer are insufficient.
Answer 6: Thank you for kindly suggesting an analysis of the chitosan, pristine modificated chitosan (ChitP) and the final product (ChitPNi). In section 3.1: was introduced 13C NMR for modification of chitosan by the 1-phosphonopropyl group. The characteristic chemical shifts for ChitP in 13C NMR appear at 18.41, 22.10, 55.77, 77.56, 59.93, 70.00, 74.73, 76.27 and 97.48 ppm and assigned to CH3, CH2, C6, C2, CH, C5, C3, C4 and C1 carbons, respectively.
- Line 103: What is the role of urea in this process?
Answer 7: Thank you for your question.
As shown in the manuscript the pH is playing an important role in the ChitPNi formation. The urea was used to control the solution pH. Under the reaction conditions, urea is slowly hydrolyzed with the release of NH3 and allowing the pH raising in a homogeneous manner.
- Section 3.1. Adsorbent characterization: Characterization of the ChitP sample using IR spectroscopy only is obviously insufficient. It is necessary to prove the modification of chitosan by the 1-phosphonopropyl group (e.g., by 1H, 13C NMR), to determine the degree of substitution (e.g., by 1H NMR), the degree of acetylation (e.g., by 1H NMR), and the molecular mass of ChitP (e.g., by SEC-MALS).
Answer 8: Thank you for your kindly suggesting. 13C NMR and 31P NMR chemical shifts were presented in the paper. Unfortunately, the NMR were done in other University and due to pandemic situation we do not have access there anymore and we cannot do the NMR for the pristine chitosan sample. In this respect we performed XPS and 13C NMR and introduced the following paragraphs in 2.4 Characterization section as following:
X-ray photoelectron spectroscopy (XPS) was performed on a KRATOSAxisNova (KratosAnalytical, Manchester, UK), using AlK radiation, with a current of 20 mA and a voltage of 15 kV (300 W). The XPS spectrum for the tested sample was made in the binding energy range -10 ÷ 1200 eV, using pass energy of 160 eV with a resolution of 1 eV. The XPS spectrum for all identified elements (N, P, O, C, Ni) was performed using a pass energy of 20 eV and a step size of 0.1 eV. Vision Processing software (Vision2 software, Version 2.2.10) was used for data analysis.
The characteristic chemical shifts for ChitP in 13C NMR appear at 18.41, 22.10, 55.77, 77.56, 59.93, 70.00, 74.73, 76.27 and 97.48 ppm and assigned to CH3, CH2, C6, C2, CH, C5, C3, C4 and C1 carbons, respectively.
This figure (Figure 3. The wide scan XPS for the ChitPNi sample) and the paragraph with references 39, 40, 41 and 42 have been introduced in 3.1. Adsorbent characterization.
Figure 3. The wide scan XPS for the ChitPNi sample.
“X-ray photoelectron spectroscopy (XPS) method was described in an previous article [39] In the analysis of the XPS spectrum in Fig. 3, we obtained elements: C, N, O P and Ni. From XPS spectrum was observed that ChitPNi exhibit for phosphorus and Ni ions comparable with the one presented in the literature [40, 41, 42]. Also, the values for the phosphorus, nitrogen and nickel content are given in Figure 3.”
- 39. Lupa, L., Cocheci, L., Trica, B., Coroaba, A., Popa, A., Photodegradation of Phenolic Compounds from Water in the Presence of a Pd-Containing Exhausted Adsorbent, Sci. 2020, 10, 8440, doi:10.3390/app10238440
- De Jesus, J. C.; Gonzalez, I.; Quevedo, A.; Puerta, T. Termal decomposition of nickel acetate tetrahydrate: an integrated study by TGA, QMS and XPS techniques. J. Mol. Catal. 2005, 228, 283-291. https://doi.org/10.1016/j.molcata.2004.09.065.
- 41. Asmaa S. Morshedy, Ahmed A. Galhoum, Abdel Aleem H. Abdel Aleem, Mohamed T. Shehab El-din , Dina M. Okaba, Mohsen.S. Mostafa, Hamed I. Mira, Zhen Yang, Ibrahim E.T.- El-Sayed, Functionalized aminophosphonate chitosan-magnetic nanocomposites for Cd(II) removal from aqueous solutions: Performance and mechanisms of sorption, Applied Surface Science. 2021, 561, 150069, 10.1016/j.apsusc.2021.150069
- B.S. Rathore, N.P.S. Chauhan, S. Jadoun, S.C. Ameta, R. Ameta, Synthesis and characterization of chitosan-polyaniline-nickel(II) oxide nanocomposite, J. Mol. Struct. 2021, 1242, 130750, 10.1016/j.molstruc.2021.130750
Acknowledgments: Dr. Adina Coroaba ([email protected]) - for the XPS survey spectra of ChitPNi sample, that was tested at Centre of Advanced Research in Bionanoconjugates and Biopolymers, “Petru Poni” Institute of Macromolecular Chemistry Iasi, Grigore Ghica Voda Alley, No.41 A, 700487 Iasi, Romania.
- The quality of FTIR spectra (Fig. 1) is poor as well as the description and discussion of the spectra (lines 242-254) are very confusing and must be revised entirely. Incorrect assignment of absorption bands (e.g., C-O from acetylated moieties of chitosan at 1350-1300 cm-1). The possible presence of acetate is not taken into account (although they mention the presence of acetic acid in the samples when describing the TG/DTA experiments, line 283).
Answer 9: Thank you for your question. The paragraph is inserted:
“The recorded FTIR spectrum of (ChitPNi) pH5 (see Figure 1) indicates the presence of a 1050 cm-1 band assigned to the –P-O-H group. The N-H stretch strips and band appear at 1655 and 1560 cm-1. A strong absorption is observed at 3430 cm-1, corresponding to the tensile vibration of the hydroxyl groups in the water coordinated with the Ni atom superimposed with the tensile vibration N-H. At 1420 cm-1 it was assigned a band for OH bending vibration. The band at 1540 cm-1 is due to vibrations of the carboxylate ion. The absence of bands in the range 2700-2560 cm-1, corresponding to the P-O vibration of the P-OH acid groups, confirms the formation (ChitPNi) [35, 36]. These dates are in line with the proposed structures of functionalized chitosan (ChitPNi).”
- Lines 254-256: How did the authors reach this conclusion without examining the degree of crystallinity?
Answer 10: Thank you for your question. We have examined ChitPNi sample using X-ray photoelectron spectroscopy (XPS). Also, the RX-diffractograms where recorded and the results were introduces in the manuscripts.
- Line 258: What do the authors mean by “31P functionalities”?
Answer 11: The paragraph was modified as following: The 31P-NMR spectrum of the raw material (ChitP) displayed only one signal at 2.858 ppm. This peak is generally expected for phosphonic groups.
- Line 298: How can NiO be removed at 320 °C as its melting point is 1.955 °C?
Answer 12: Thank you for your question. In fact, the process described in the paragraph below takes place:
At 320 oC, NiO is obtained [36, 40], and at 358 oC, P2O5 is detached [43]. The mass loss of the oxidative degradation step was 42.62% and the total loss of the (ChiPNi) sample is 56.21%.
- Lines 397-400: Does nickel release during the regeneration of 0.1 M HCl adsorbent? If not, what would be the reason?
Answer 13: Thank you for your question. The nickel ions are not released during the regeneration with 0,1 M HCl, due to the fact that between the Ni ions and ChitP are formed a complex, according to the RX and XPS analysis, their interactions are more strong, is not just a physical sorption like in case of lead ions which is absorbed on the materials surface.
- The discussion of the results of the paper is also rather weak. I still do not comprehend the role of nickel ions in the adsorbent. The authors claimed that there is no competitive sorption of lead, leading to displacement of nickel from the sorbent. If so, maybe nickel is not needed in the sorbent at all? To resolve this question, I would like to see in the paper a sorption experiment performed on a nickel-free sample (ChitP).
Answer 14: The equilibrium isotherm of Pb ions adsorption onto nickel-free sample, ChitP was introduced in figure 11. It could be observed that through modification of ChitP sample with Ni ions is obtained an adsorbent material with a higher active sites, which lead to the development of a doubling maximum adsorption capacity compared with the adsorption capacity developed by ChitP sample.

Reviewer 2 Report
Dear Authors,
The topic of the manuscript is very interesting, novel and it provides new information to the scientific field. I feel that this work will be attractive in the community. However, I propose the following changes for the improvement of the manuscript quality.
Please add some brief information about other natural sorbents (and modified one) used for lead ions sorption/remove;
Please add some references to sentence in lines 54-56 - about different forms of chitosan in sorption;
Line 35, 36, 82, 83... - check the name of compounds, according to IUPAC nomenclature of organic compounds: N - should be written in italics, and D - as a capital letter;
Please check all units in the manuscript: ex. Line 95 and 97 - various symbol of degree - use one type in whole manuscript or Line 123: mg mL-1 (-1) should be superscript;
Please check all shortcuts - e.g. choose ChitPNi5 or ChitPNipH5 - not interchangeably;
Figures 2 or 6 - Please change / correct it - very low, poor quality making it difficult to interpret;
Heavy metal sorption on chitosan is pH dependent, so why only results for pH5 are shown in this paper;
Table 3. - Maybe use qeexp instead qeexp, like qecalc;
Please expand the Conclusions section.
All things considered, this manuscript addresses an important research topic, however there are some mistakes, which should be corrected. Therefore, I suggest the major revision of this article.
Author Response
Dear Reviewer thank you for your suggestion and comments, we improved our manuscript according with you recommendation and we hope that in this format this could be accepted for publication.
The topic of the manuscript is very interesting, novel and it provides new information to the scientific field. I feel that this work will be attractive in the community. However, I propose the following changes for the improvement of the manuscript quality.
- Please add some brief information about other natural sorbents (and modified one) used for lead ions sorption/remove;
Answer 1: Thank you for your suggestion. The paragraph with the references [15-17] about the sorption of Pb ions was introduced
“Natural polysaccharide-type absorbents (eg chitosan or cellulose) are used to remove the Pb (II) ion, which has been shown to have good sorption properties [15-17]. Pollution of lead ions in wastewater is a serious problem. Therefore, we examined the ability of ChitPNi as an adsorbent for lead ions, expecting good adsorption properties.”
- Jin, X., Xiang, Z., Liu, Q., Chen, Y., Lu, F., Polyethyleneimine-bacterial cellulose bioadsorbent for effective removal of copper and lead ions from aqueous solution, Bioresour. Technol. 2017, 244, 844-849, doi:10.1016/j.biortech.2017.08.072
- Jiao Lu, Ru-Na Jin, Chao Liu, Yan-Fei Wang, Xiao-kun Ouyang, Magnetic carboxylated cellulose nanocrystals as adsorbent for the removal of Pb(II) from aqueous solution, Int. J. Biol. Macromol. 2016, 93, 547-556 doi:10.1016/j.ijbiomac.2016.09.004 0141-8130
- Kumar, R., Sharma, R.Kr., Synthesis and characterization of cellulose based adsorbents for removal of Ni(II), Cu(II) and Pb(II) ions from aqueous solutions, React. Funct. Polym. 2019, 140, 82-92, doi:10.1016/j.reactfunctpolym.2019.04.014
- Please add some references to sentence in lines 54-56 - about different forms of chitosan in sorption;
Answer 2: Thank you for your suggestion. On Line 54-55 the two references are placed [13-17]
“Chitosan and products derived from it have presented in treatment processes of water for the elimination of metal ions such as: zinc (II), copper (II), cadmium (II), nickel (II), lead (III), lead (II) ions [2,11,12] and for the removal of fluoride from waste waters [13-17]. The potential of chitosan in adsorption of heavy metals can be linked to: the high hydrophilicity on glucose units due to the high number of hydroxyl groups, the high reactivity of the additional functional groups and the flexible structure of the polymer chain [10].”
- Cho, Dong-Wan; Jeon, Byong-Hun; Jeong, Yoojin; Nam, In-Hyun; Choi, Ui-Kyu; Kumar, Rahul; Song, Hocheol, Synthesis of Hydrous Zirconium Oxide-Impregnated Chitosan Beads and Their Application for Removal of Fluoride and Lead, Appl. Surf. Sci. 2016, 372, doi:10.1016/j.apsusc.2016.03.068
- Salehi, E., Daraei, P., Shamsabadi, A.A., A review on chitosan-based adsorptive membranes, Carbohydr. Polym. 2016, 152, 419-432, doi:10.1016/j.carbpol.2016.07.033
- Jin, X., Xiang, Z., Liu, Q., Chen, Y., Lu, F., Polyethyleneimine-bacterial cellulose bioadsorbent for effective removal of copper and lead ions from aqueous solution, Bioresour. Technol. 2017, 244, 844-849, doi:10.1016/j.biortech.2017.08.072
- Jiao Lu, Ru-Na Jin, Chao Liu, Yan-Fei Wang, Xiao-kun Ouyang, Magnetic carboxylated cellulose nanocrystals as adsorbent for the removal of Pb(II) from aqueous solution, Int. J. Biol. Macromol. 2016, 93, 547-556 doi:10.1016/j.ijbiomac.2016.09.004 0141-8130
- Kumar, R., Sharma, R.Kr., Synthesis and characterization of cellulose based adsorbents for removal of Ni(II), Cu(II) and Pb(II) ions from aqueous solutions, React. Funct. Polym. 2019, 140, 82-92, doi:10.1016/j.reactfunctpolym.2019.04.014
- Line 35, 36, 82, 83... - check the name of compounds, according to IUPAC nomenclature of organic compounds: N- should be written in italics, and D - as a capital letter;
Answer 3: Thank you for your suggestion. The modifications were performed.
- Please check all units in the manuscript: ex. Line 95 and 97 - various symbol of degree - use one type in whole manuscript or Line 123: mg mL-1 (-1) should be superscript;
Answer 4: Thank you for your suggestion. The modifications and corrections were performed in whole document.
- Please check all shortcuts - e.g. choose ChitPNi5 or ChitPNipH5 - not interchangeably;
Answer 5: Thank you for your suggestion. The ChiPNi was chosen. In order to indicate the pH we used the following notation ChitPNi (pH5) in all manuscript.
- Figures 2 or 6 - Please change / correct it - very low, poor quality making it difficult to interpret;
Answer 6: Thank you for your suggestion. The resolution of the figures 2 and 6 was improved.
- Heavy metal sorption on chitosan is pH dependent, so why only results for pH5 are shown in this paper;
Answer 7: Thank you for you observation. It was a mistake of expression in section 3.4. In the manuscript we didn’t used Pb solution having different initial pH. The adsorption process were conducted only at pH=5, as it was mentioned in the manuscript in section 2.5. See the paragraph bellow.
The aqueous solutions containing different amounts of lead ions were obtained through adequate dilution of the stock solution using distilled water. During the adsorption process the initial pH of the lead containing aqueous solutions was adjusted around the value pH=5. At this pH the obtained modified adsorbent presents the best characteristics, and also according to a literature survey the best pH value to use in the adsorption of lead ions onto various chitosan derivatives is in the range: 4.5-5.5
And at section 3.4. we removed the paragraph regarding the pH influence. The studied adsorbent materials presented the highest stability at this pH, therefore all the adsorption process were conducted at pH=5.
- Table 3. - Maybe use qeexp instead qeexp,like qecalc;
Answer 8: Thank you for you observation, we made the changes in Table 3.
- Please expand the Conclusions section.
Answer 9: The conclusion section was expand
All things considered, this manuscript addresses an important research topic, however there are some mistakes, which should be corrected. Therefore, I suggest the major revision of this article.

Round 2
Reviewer 1 Report
The authors have successfully addressed most of the reviewers’ concerns in the revised manuscript and have made the necessary revisions. As a result, the paper has been generally improved and, in my opinion, can be published in the present form.
Reviewer 2 Report
The authors tried to improve the manuscript, but did not address all of the reviewer's suggestions, for example, concerning the naming of organic compounds such as D-glucosamine - capital letter for D isomer; they did not use the correct styles in the descriptions of tables and figures...
Nevertheless, I request that this manuscript be accepted